# IMITATION LEARNING FROM OBSERVATIONS UNDER TRANSITION MODEL DISPARITY

## ABSTRACT

Learning to perform tasks by leveraging a dataset of expert observations, also known as imitation learning from observations (ILO), is an important paradigm for learning skills without access to the expert reward function or the expert actions. We consider ILO in the setting where the expert and the learner agents operate in different environments, with the source of the discrepancy being the transition dynamics model. Recent methods for scalable ILO utilize adversarial learning to match the state-transition distributions of the expert and the learner, an approach that becomes challenging when the dynamics are dissimilar. In this work, we propose an algorithm that trains an intermediary policy in the learner environment and uses it as a surrogate expert for the learner. The intermediary policy is learned such that the state transitions generated by it are close to the state transitions in the expert dataset. To derive a practical and scalable algorithm, we employ concepts from prior work on estimating the support of a probability distribution. Experiments using MuJoCo locomotion tasks highlight that our method compares favorably to the baselines for ILO with transition dynamics mismatch.

## 1 INTRODUCTION

Imitation Learning (IL) is a framework that trains an agent to perform desired skills by leveraging expert demonstrations of those skills. Compared to the standard Reinforcement Learning (RL) approach, IL offers the benefit of not requiring a reward function, that can be difficult to specify for complicated objectives. Recent IL methods that integrate efficiency with deep-RL and are performant in high-dimensional state-action spaces include behavioral-cloning-based algorithms (Ross et al., 2011; Brantley et al., 2019), and adversarial IL algorithms inspired by maximum entropy inverse-RL (Ho & Ermon, 2016; Finn et al., 2016). Imitation Learning from Observations (ILO) refers to the setting where the expert demonstrations consist of only observations (or states), while the expert actions are unavailable. ILO is beneficial when the measurement of the expert action is difficult, *e.g.*, in kinesthetic teaching in robotics or when learning with motion capture datasets.

Adversarial IL methods frame the problem as the minimization of an *f*-divergence between the state-action visitation distributions of the expert and the learner (Ke et al., 2019). Since expert actions are absent in ILO, the analogous methodology here is to minimize the *f*-divergence between the state-transition distributions of the expert and the learner (Arumugam et al., 2020). A state-transition distribution for a policy is the joint distribution over the current state and the next state, and is defined formally in Section 2. Choosing the *f*-divergence to be the Jensen–Shannon (JS) divergence has enabled successful imitation using algorithms such as GAIL (Ho & Ermon, 2016) and GAIfO (Torabi et al., 2018b), for IL and ILO, respectively.

The transition dynamics model of an environment governs the distribution over the next state, given the current state and action. In this paper, we focus on ILO in the scenario of a transition dynamics mismatch between the expert and the learner environments. Such discrepancy could manifest in real-world applications of imitation learning when there are subtle differences in the physical attributes of the system used to collect the demonstrations and the system where the learner policy is run. Adversarial ILO methods such as GAIfO, that attempt to train the learner by matching its state-transition distribution with that of the expert, perform very well when the expert and learner operate in a shared environment, under the same dynamics. When the dynamics differ, however, matching the state-transition distributions becomes challenging since the state transitions provided in the ex-

pert demonstrations could be infeasible under the dynamics in the learner's environment. To form some intuition, consider ILO in a 2D grid with discrete states, where the path demonstrated by the expert follows the main diagonal from the bottom-left grid-cell to the top-right grid-cell. Suppose that the transition dynamics for the learner environment are such that only horizontal and vertical moves are permitted on the grid, *i.e.,* no diagonal motion for the learner. In this case, an optimal learner's movement is still in the same general direction as the expert, and it covers all the expert states (along with some adjacent states). However, matching the state-transition distributions is an ineffective strategy since the learner cannot reproduce the (one-step) state transitions of the expert.

To alleviate the challenges of ILO under dynamics mismatch, we propose an algorithm that trains an intermediary policy in the learner environment, and hope to use it as a surrogate expert for training the learner (imitator). We refer to this policy as the *advisor*. For the advisor to be effective, the state transitions generated by it in the learner environment should be as close as possible to the state transitions in the expert dataset. We formalize this concept in terms of the cross-entropy distance between state-conditional next-state distributions of the expert and the advisor. To convert this into a practical and scalable algorithm for training the advisor, we incorporate ideas from distribution support estimation (Wang et al., 2019). Simultaneous to the advisor training, the learner agent is updated to imitate the advisor. Crucially, the advisor operates in the same environment as the learner, making the distribution-matching IL objective amenable.

We evaluate the efficacy of our ILO algorithm using five locomotion control tasks from OpenAI Gym where we introduce a mismatch between the dynamics of the expert and the learner by changing different configuration parameters. We demonstrate that our approach compares favorably to the baseline ILO algorithms in many of the considered scenarios.

## 2 PRELIMINARIES

We model the RL environment as a discounted, infinite horizon Markov Decision Process (MDP). At every discrete timestep, the agent observes a state ($s \in \mathcal{S}$), generates an action ($a \in \mathcal{A}$) from a stochastic policy $\pi(a|s)$, receives a scalar reward $r(s, a)$, and transitions to the next state ($s'$) sampled from the transition dynamics model $p(s'|s, a)$. In the infinite horizon setting, the future rewards are discounted by a factor of $\gamma \in [0, 1)$. Let $d_\pi^t(s)$ denote the distribution induced by $\pi$ over the state-space at a particular timestep $t$. The stationary discounted state distribution of $\pi$ is then defined as $\rho_\pi(s) = (1 - \gamma) \sum_{t=0}^\infty \gamma^t d_\pi^t(s)$. The RL objective of maximizing the cumulative discounted sum of rewards can be framed as $\max_\pi \mathbb{E}_{\rho_\pi(s,a)}[r(s, a)]$, where $\rho_\pi(s, a) = \rho_\pi(s)\pi(a|s)$ is the state-action distribution (also known as the occupancy measure). Lastly, we define the state-transition distribution for a policy as $\rho_\pi(s, s') = \sum_a p(s'|s, a)\rho_\pi(s, a)$ [1].

### 2.1 GAIL AND GAIFO

Generative Adversarial Imitation Learning (Ho & Ermon, 2016) is a widely popular model-free IL method that builds on the Maximum Causal Entropy Inverse-RL (MaxEnt-IRL) framework (Ziebart, 2010). MaxEnt-IRL models the expert behavior with a policy that maximizes its $\gamma$-discounted causal entropy, $\mathcal{H}(\pi) = \mathbb{E}_\pi[-\log \pi(a|s)]$, while satisfying a feature matching constraint. GAIL considers a regularized version of the dual to this primal problem. It shows that RL with the reward function recovered as the solution of the regularized dual is equivalent to directly learning a policy whose state-action distribution is similar to that of the expert. For a specific choice of the regularizer, this similarity is quantified by the JS divergence between the two state-action distributions, $D_{JS}[\rho_\pi(s, a) \,||\, \rho_{\pi_e}(s, a)]$. Based on these ideas, GAIL seeks to learn a policy with the objective:

$$\min_\pi \max_D \mathbb{E}_{\rho_\pi}\big[\log D(s, a)\big] + \mathbb{E}_{\rho_{\pi_e}}\big[\log(1 - D(s, a))\big] - \lambda \mathcal{H}(\pi)$$

where $D : \mathcal{S} \times \mathcal{A} \to (0, 1)$ is the discriminator that provides the rewards for training the learner policy $\pi$, and the inner maximization over $D$ approximates $D_{JS}(\cdot)$ similar to GANs (Goodfellow et al., 2014). To empirically estimate the expectation under $\rho_{\pi_e}(s, a)$, state-action pairs are sampled from the available expert demonstrations. In the ILO setting, however, expert actions are not included in the demonstrations. To mitigate this challenge, Torabi et al. (2018b) propose GAIfO, which adapts

---

[1]With slight abuse of notation, we use the symbol $\rho_\pi$ for state, state-action, and state-transition distributions of a policy $\pi$. We would provide context around their usage to avoid any confusion.

to ILO by modifying the GAIL objective to match the *state-transition* distributions of the expert and the learner, *i.e.,* $D_{JS}[\rho_\pi(s, s') \| \rho_{\pi_e}(s, s')]$. Correspondingly, the discriminator in GAIfO is a function of the state transitions $D(s, s')$.

## 2.2 SUPPORT ESTIMATION VIA RED

We summarize a recently proposed method for estimating the support of a distribution in high dimensions (RED; Wang et al. (2019)), since it forms a core ingredient of our final algorithm. Let $\mathcal{X}$ denote a set and $p$ be a probability distribution on $\mathcal{X}$. Denote by $\text{supp}(p) = \{x \in \mathcal{X} \mid p(x) \neq 0\}$, the support of the distribution $p$. Given any $x \in \mathcal{X}$, the task is to know if $x \in \text{supp}(p)$. Towards this goal, Wang et al. (2019) combine ideas from kernel-based support estimation (De Vito et al., 2014) and RND (Burda et al., 2018), and consider the following objective:

$$\theta^* = \arg\min_\theta \mathbb{E}_{x \sim p(x)} \| f_\theta(x) - f_{\tilde{\theta}}(x) \|_2^2$$

where $f_\theta : \mathcal{X} \to \mathbb{R}^K$ is a trainable function parameterized by $\theta$, while $f_{\tilde{\theta}}$ is a *fixed* function with randomly initialized parameters $\tilde{\theta}$. Define the score function as (with constant positive scalar $\lambda$):

$$r_{\text{RED}}(x) = \exp(-\lambda \| f_{\theta^*}(x) - f_{\tilde{\theta}}(x) \|_2^2)$$

Wang et al. (2019) conclude that the score $r_{\text{RED}}(x)$ is high if $x \in \text{supp}(p)$, and is low otherwise. With neural network function approximators, we thus obtain a smooth metric whose value decreases (increases) as we move farther from (closer to) the support of the distribution $p$.

## 3 METHODS

We begin by defining some notations for our setup. We denote the MDP in which the expert policy ($\pi_e$) operates as the $e$-MDP, while the learner (or imitator) policy is run in the $l$-MDP. The two MDPs share all the attributes, except for the transition dynamics function, which we symbolize with $p_e(s'|s, a)$ and $p_l(s'|s, a)$, for the $e$-MDP and $l$-MDP, respectively. $\rho_e(s)$, $\rho_e(s, a)$, and $\rho_e(s, s')$ are the state, state-action, and state-transition distribution for the expert policy. These distributions depend on the $e$-MDP dynamics $p_e(s'|s, a)$. The corresponding distributions for the learner are denoted by $\rho_\pi(\cdot)$ and they depend on the $l$-MDP dynamics $p_l(s'|s, a)$.

Adversarial ILO methods, such as GAIfO, that learn the imitator policy by matching the state-transition distributions of the expert and learner aim to solve the following primal problem:

$$\max_\pi \mathcal{H}(\pi) \quad \text{s.t.} \quad \rho_e(s, s') = \rho_\pi(s, s') \quad \forall (s, s') \in \mathcal{S} \times \mathcal{S} \tag{1}$$

If the expert and the learner operate under different transition dynamics, depending on the extent of the mismatch, it is possible that some (or all) of the one-step state transitions of the expert are infeasible under the dynamics function in $l$-MDP. Said differently, given a $(s_e, s'_e)$ pair sampled from the expert demonstrations, there could be no action in the $l$-MDP from the state $s_e$ that results in $s'_e$ as the next state, as per the dynamics $p_l(s'|s, a)$. This renders the state-transition matching objective hard to optimize in practice. For instance, in the practical implementation of GAIfO, the rewards for the imitator policy are computed from a discriminator that is trained with binary classification on $(s, s')$ pairs from the expert and the learner. If the expert transitions can't be generated in $l$-MDP by the learner, then a high-capacity discriminator could achieve perfect accuracy and thus fail to provide informative rewards for imitation.

Consider a sequence of states $\{s_1, s_2, \dots\}$ generated by the expert policy in the $e$-MDP, as shown in Figure 1a. Given an expert state $s_i$, it may not be possible to reach $s_{i+1}$ in a single timestep in the $l$-MDP. Instead, we would like to find an alternative state $\tilde{s}_{i+1}$ that *is* reachable from $s_i$ in one step (*i.e.,* feasible under the dynamics $p_l$) and is close to the desired destination $s_{i+1}$. For instance, in Figure 1a, starting from the expert state $s_1$, $\tilde{s}_2$ is a more desirable next state compared to $\hat{s}_2$.

### 3.1 GUIDANCE VIA AN ADVISOR POLICY

To discover such feasible states $\tilde{s}_i$, we introduce an advisor policy $\pi_a$ that operates in the $l$-MDP. $\pi_a$ is invoked only for action selection on the expert states, rather than being run in a closed feedback

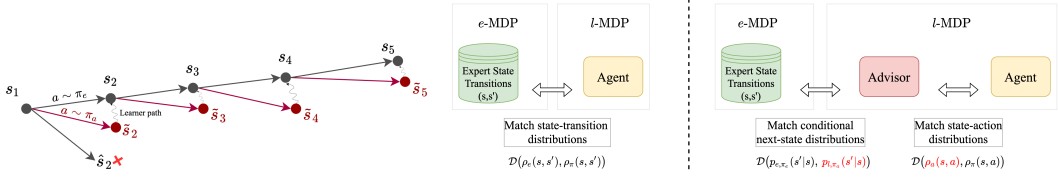

(a) State transitions        (b) Comparison between GAIfO (left) and our approach (right)

Figure 1: (a) A sequence of states $s_i$ from the expert dataset. The states $\tilde{s}_i$ are reached by sampling an action from the advisor policy $\pi_a$ from every expert state. State $\hat{s}_2$ is less desirable than $\tilde{s}_2$ since the latter is closer to the expert state $s_2$. The squiggly lines show the path that a learner, that is optimized to match the state-action distribution of the advisor, may take. (b) A high-level overview of our approach. While GAIfO directly matches the state-transition distributions of the expert and the learner, we learn an intermediary policy (advisor) in the $l$-MDP that acts as the surrogate expert for the learner. $\mathcal{D}$ is used to denote the corresponding distance metric.

loop in the learner environment. In this way, $\pi_a$ is akin to a contextual bandit policy. The goal with $\pi_a$ is to produce an action $a_i \sim \pi_a(\cdot|s_i)$ from the expert state $s_i$ such that the next state in the $l$-MDP, $\tilde{s}_{i+1} \sim p_l(\cdot|s_i, a_i)$, is close to the next state $s_{i+1}$ in the $e$-MDP under the expert policy. We expand on the measure of *closeness* and the methodology to train the advisor in the next subsection.

When the expert and learner dynamics are the same, the optimal advisor would take the same actions as the expert policy. In the presence of a dynamics mismatch, however, the advisor actions provide reasonably good guidance on how to stay close to the expert's state trajectory (Figure 1a). Therefore, the advisor $\pi_a$ could act as a *surrogate expert* for the learner and the learner's imitation learning objective could be suitably modified. The main advantage of the learner and the advisor operating in the same $l$-MDP is that we could now attempt to match their visitation distributions, which is much more structured than matching distributions across dynamics (Eq. 1). We consider the IL objective:

$$\max_{\pi} \ \mathcal{H}(\pi) \quad \text{s.t.} \quad \rho_{\pi_a}(s,a) = \rho_{\pi}(s,a) \quad \forall \, (s,a) \in \mathcal{S} \times \mathcal{A}$$

where $\rho_{\pi_a}(s,a)$ is the state-action distribution of $\pi_a$. Since the advisor is invoked only on the expert states, effectively, $\rho_{\pi_a}(s,a) = \rho_e(s)\pi_a(a|s)$. We can convert the above objective to an unconstrained optimization by using a parametric function $f_\omega(s,a)$ as the Lagrange multiplier:

$$\min_{\omega} \max_{\pi} \ \mathcal{J}(\pi, \omega) := \mathcal{H}(\pi) + \mathbb{E}_{\rho_e(s)\pi_a(a|s)}[f_\omega(s,a)] - \mathbb{E}_{\rho_\pi(s,a)}[f_\omega(s,a)] \tag{2}$$

We note that this objective bears resemblance to entropy-regularized apprenticeship learning (Abbeel & Ng, 2004; Syed et al., 2008; Syed & Schapire, 2008), modulo the use of an advisor policy that contributes the favorable actions, instead of the expert policy.

## 3.2 TRAINING THE ADVISOR

The advisor policy generates an action (deterministic $\pi_a$), or a distribution over actions (stochastic $\pi_a$), given an expert state sampled from the demonstration data. One suitable objective for training the advisor is to minimize the dissimilarity between the destination state achieved with the expert policy in $e$-MDP and that achieved with the advisor in $l$-MDP. For instance, consider the scenario where the dynamics functions are deterministic and known, denoted by $f_e$ and $f_l$ for the $e$-MDP and $l$-MDP, respectively. Also, the policies $\pi_e$ and $\pi_a$ are deterministic. Then, the advisor could be optimized with the following loss:

$$\min_{\pi_a} \ \mathcal{J}(\pi_a) := \mathbb{E}_{s \sim \rho_e(s)}\Big[\mathcal{D}\big[f_e(s, \pi_e(s)), f_l(s, \pi_a(s))\big]\Big]$$

where $\mathcal{D}$ is a distance measure, *e.g.*, the $L_2$-norm in the state space. However, we are interested in the setup with stochastic, unknown dynamics functions and stochastic policies. To compute a distance metric in this setting, we first define the state-conditional next-state distributions for the expert and the advisor, by marginalizing over the actions:

$$p_{e,\pi_e}(s'|s) = \sum_a p_e(s'|s,a)\pi_e(a|s) \quad ; \quad p_{l,\pi_a}(s'|s) = \sum_a p_l(s'|s,a)\pi_a(a|s)$$

We then use the cross-entropy distance ($\mathbb{H}$) between these distributions as a measure of closeness between the expert and the advisor:

$$
\begin{aligned}
\pi_a^* &= \underset{\pi_a}{\arg\min} \ \mathcal{J}(\pi_a) \coloneqq \mathbb{E}_{s \sim \rho_e(s)} \Big[ \mathbb{H} \big[ p_{l,\pi_a}(s'|s), \ p_{e,\pi_e}(s'|s) \big] \Big] \\
&= \underset{\pi_a}{\arg\max} \ \mathcal{J}(\pi_a) \coloneqq \mathbb{E}_{s \sim \rho_e(s)} \mathbb{E}_{a \sim \pi_a(a|s)} \mathbb{E}_{s' \sim p_l(s'|s,a)} \big[ \log p_{e,\pi_e}(s'|s) \big] \\
&= \underset{\pi_a}{\arg\max} \ \mathcal{J}(\pi_a) \coloneqq \mathbb{E}_{s \sim \rho_e(s)} \mathbb{E}_{a \sim \pi_a(a|s)} \mathbb{E}_{s' \sim p_l(s'|s,a)} \big[ \log \rho_e(s,s') \big]
\end{aligned}
\tag{3}
$$

where, in the last equation, we have multiplied the term inside the log with $\rho_e(s)$, a quantity independent of $\pi_a$. Figure 1b provides a high-level overview of our approach.

### 3.2.1 AN APPROXIMATION BASED ON RED

The objective in Eq. 3 presents a couple of challenges. Firstly, it is infeasible to evaluate the density of any state transition $(s,s')$ under the expert's state-transition distribution $\rho_e(s,s')$, since this distribution is unknown and only a few samples from this distribution are available to us in the form of the expert demonstrations. Secondly, and more importantly, note that the state transition that ought to be evaluated under $\rho_e(s,s')$ is generated by the advisor policy in the $l$-MDP. If no advisor policy can replicate the state transition behavior of the expert, as is likely when there is a dynamics mismatch, then the objective becomes degenerate with an optimal value of $-\infty$.

Our approach to mitigate these issues is to replace the log-density term in Eq. 3 with an estimated value that quantifies the *proximity* of a given state transition $(s,s')$ to the manifold of the expert's support in this space, *i.e.*, $\rho_e(s,s')$. To get this value, we leverage RED (Wang et al., 2019), which pre-trains a deep neural network using data samples from the distribution of interest. Then, given a test sample, the network outputs a continuous value that provides an estimate of how far the test sample is from the distribution's support. Section 2.2 provides a short background on RED.

In our instantiation, we pre-train a RED network $r_{\text{RED}}^{\phi}(s,s')$ with state transitions from the expert demonstrations. The network is then frozen and utilized in the objective to train the advisor:

$$
\max_{\pi_a} \ \mathcal{J}(\pi_a) \coloneqq \mathbb{E}_{s \sim \rho_e(s)} \mathbb{E}_{a \sim \pi_a(a|s)} \mathbb{E}_{s' \sim p_l(s'|s,a)} \big[ r_{\text{RED}}^{\phi}(s,s') \big]
\tag{4}
$$

For any advisor-generated state transition, $r_{\text{RED}}^{\phi}(s,s')$ provides a smooth metric in the range $(0,1]$, whose value increases (decreases) as the transition moves closer to (farther from) the support of the distribution $\rho_e(s,s')$. Thus, training $\pi_a$ to maximize this value yields an advisor that provides guidance on how to stay close to the expert's state trajectory when operating in $l$-MDP (Figure 1b).

To provide further intuition on the use of RED, Table 1 shows the $r_{\text{RED}}^{\phi}(s,s')$ values obtained from a trained RED network in two environments—*Walker2d* and *HalfCheetah*. The RED network is trained with 50 state transitions $\{s_e, s'_e\}$ sampled from $\rho_e(s,s')$ and then evaluated under several noisy transitions $\{s_e, s'_e + \eta\}$, where $\eta$ denotes zero-mean Gaussian noise. We show the $r_{\text{RED}}^{\phi}$ values averaged across the transitions for different settings of the noise-to-signal ratio (N/S), *i.e.*, the ratio of the standard deviation of the noise to the standard deviation of the states. We observe that $r_{\text{RED}}^{\phi}(s,s')$ is close to 1.0 when the transitions are on the support of the expert, and it gradually decreases as the transitions drift away from the support as a result of a larger amount of added noise.

| Increasing noise ↓ | Walker2d | HalfCheetah |
|---|---|---|
| N/S = 0 | 0.93 | 0.96 |
| N/S = 0.2 | 0.80 | 0.71 |
| N/S = 0.5 | 0.44 | 0.23 |
| N/S = 1 | 0.12 | 0.04 |
| N/S = 2 | 0.01 | 0.001 |

Table 1: Averaged $r_{\text{RED}}^{\phi}(s,s')$ values for different noise levels.

### 3.3 ALGORITHM AND IMPLEMENTATION

It is possible to use a two-stage training procedure for the overall algorithm—first, learn the advisor in $l$-MDP with Eq. 4, and then use it in the IL objective laid out in Eq. 2 to optimize for the reward network $f_\omega$ and the learner policy $\pi$. In the first stage of advisor learning, although the RED network is trained offline, computing the gradients for optimizing $\pi_a$ still requires environment interaction. More importantly, since the advisor is only trained on the expert states, the objective in Eq. 4 requires the capability to reset the environment to the expert states in the $l$-MDP.

---

**Algorithm 1:** AILO (Advisor-augmented Imitation Learning from Observations)

---

**Input:** A dataset of expert state transitions $\mathcal{D}_e = \{s_e, s'_e\}$ collected in $e$-MDP

/* initialization and offline pre-training */
Initialize: advisor $\pi_a$, learner $\pi$, reward network $f_\omega(s, a)$, RED network $r^\phi_{\text{RED}}(s, s')$
Pre-train $r^\phi_{\text{RED}}(s, s')$ with $\mathcal{D}_e$ using the algorithm in Wang et al. (2019)

**for** *iter in* $\{1, \dots, N\}$ **do**

    /* data collection */
    Roll out trajectories $\tau$ using $\pi$

    Update reward network $f_\omega$ using $\tau$, $\pi_a$ and $\mathcal{D}_e$ (Eq. 2)

    /* update learner */
    Compute reward $f_\omega(s, a)$ for each transition in $\tau$. Use $\tau$ to update $\pi$ with MaxEnt RL

    /* update advisor */
    Compute reward $r^\phi_{\text{RED}}(s, s')$ for each transition in $\tau$. Use $\tau$ to update $\pi_a$ with Eq. 5
**end**

---

To alleviate this problem, we propose to train the advisor ($\pi_a$), the reward network ($f_\omega$) and the learner ($\pi$) jointly, in an iterative manner, and reuse the environment interaction data generated with $\pi$ for training $\pi_a$ as well, using the importance sampling trick. Specifically, let $\beta$ denote the parameters of $\pi_a$. Then the gradient of the objective function $\mathcal{J}(\pi_a)$ is:

$$
\begin{aligned}
\nabla &= \nabla_\beta \left( \mathbb{E}_{s \sim \rho_e(s)} \mathbb{E}_{a \sim \pi_a(a|s)} \mathbb{E}_{s' \sim p_l(s'|s,a)} \left[ r^\phi_{\text{RED}}(s, s') \right] \right) \\
&= \mathbb{E}_{s \sim \rho_e(s)} \mathbb{E}_{a \sim \pi_a(a|s)} \left( \mathbb{E}_{s' \sim p_l(s'|s,a)} [r^\phi_{\text{RED}}(s, s')] . \nabla_\beta \log \pi_a(a|s) \right) \\
&= \mathbb{E}_{\rho_\pi(s,a)} \underbrace{\frac{\rho_e(s)}{\rho_\pi(s)} \frac{\pi_a(a|s)}{\pi(a|s)}}_{\text{IS ratios}} \left( \mathbb{E}_{s' \sim p_l(s'|s,a)} [r^\phi_{\text{RED}}(s, s')] . \nabla_\beta \log \pi_a(a|s) \right)
\end{aligned}
\tag{5}
$$

where the second equation uses the score function estimator (Kleijnen & Rubinstein, 1996), and the third employs two importance sampling ratios. These ratios are easy to estimate for our setting (please see Appendix A.1 for details). Crucially, since the gradient is now computed with state-action data from $\rho_\pi(s, a)$, we no longer require environment resets. Further, we observe that approximating the inner expectation with a single sample is sufficient for our tasks.

We abbreviate our method as *AILO*, short for Advisor-augmented Imitation Learning from Observations. Algorithm 1 provides an outline. We start with the offline pre-training of the RED network using a dataset of expert state transitions collected in the $e$-MDP. Then, we perform iterative optimization in the $l$-MDP where each iteration involves generating trajectories with the current learner $\pi$, followed by gradient updates to the reward network $f_\omega$, and to the advisor and learner policies. Following Eq. 2, the learner policy is trained with a MaxEnt RL algorithm with per-timestep entropy-regularized reward given by $f_\omega(s_t, a_t) - \alpha \log \pi(a_t|s_t)$, where $\alpha$ is the entropy coefficient. In our experiments, we use the clipped-ratio PPO algorithm (Schulman et al., 2017) and adaptively tune $\alpha$ as suggested in prior work (Haarnoja et al., 2018).

## 4 RELATED WORK

There is a vast amount of literature on IL since it is a powerful framework to train agents to perform complex behaviors without a reward specification. ILO, where no expert action labels are available, presents several benefits as well as some unique challenges, and thus, has garnered significant attention from the community in recent times (Torabi et al., 2018a;b; Liu et al., 2018; Edwards et al., 2019; Sun et al., 2019; Yang et al., 2019; Zhu et al., 2021). ILO methods adapted from GAIL have been proposed for one-shot imitation of diverse behaviors (Wang et al., 2017), training policies to generate human-like movement patterns using motion-capture data (Peng et al., 2018; Merel et al.,

2017), and for locomotion control from raw visual data (Torabi et al., 2018b). Arumugam et al. (2020) introduce a framework that casts adversarial ILO as $f$-divergence minimization and provide insights on the design decisions that impact performance.

Several approaches have been proposed to handle the differences between the expert and the learner environments in terms of viewpoints, visual appearances, presence of distractors, and morphology changes (Stadie et al., 2017; Gupta et al., 2017; Liu et al., 2018; Sermanet et al., 2018). They typically proceed by learning a domain-invariant representation and matching features in that space. Learning such a representation is not required in our setup since the state-space is shared between the $l$-MDP and the $e$-MDP. Methods for robustness to shifts in the action-space and the dynamics model have also been researched. Zolna et al. (2019) propose to match state-pair distributions of the expert and the learner, where the states in a pair are sampled with random time gaps, rather than being consecutive. Liu et al. (2019) learn an inverse action model to predict deterministic actions in the $l$-MDP that could generate expert-like state transitions and use it to regularize policy updates. Gangwani & Peng (2020) filter the trajectories generated in $l$-MDP based on their similarity to the states in the expert dataset and perform (self-) imitation on these. The key methodological difference between these methods and our work is that we learn an intermediary stochastic policy (advisor) in the $l$-MDP by bringing its state-conditional next-state distribution closer to that of the expert, and propose an instantiation of this idea using an approximation based on support estimation.

## 5  EXPERIMENTS

We evaluate the efficacy of AILO using continuous-control locomotion environments from OpenAI Gym (Brockman et al., 2016), modeled using the MuJoCo physics simulator (Todorov et al., 2012). We include a description of the tasks, the baselines, and the learning curves. Further details on the architectures and the hyperparameters are provided in Appendix A.2.

**Environments.** We consider five tasks - {Half-Cheetah, Walker, Hopper, Ant, Humanoid}. To create a discrepancy between the expert and the learner transition dynamics, we modify one physical property in the learner environment from the set - {density, gravity, joint-friction}. Specifically, for a task $\mathcal{T}$ from the task-set, we denote:

- $\mathcal{T}$ (heavy) $\rightarrow$ learner agent has $2\times$ the mass of the expert agent
- $\mathcal{T}$ (light) $\rightarrow$ gravity in the learner's environment is half the value in the expert's
- $\mathcal{T}$ (drag) $\rightarrow$ the friction coefficient for all the joints in the learner is $2\times$ the value in the expert

**Baselines.** We contrast AILO with two baselines – a.) *GAIfO* strives to minimize the JS divergence between the state-transition distributions and is briefly described in §2.1; b.) *VAIL* (Peng et al., 2018) applies the concept of variational information bottleneck (Alemi et al., 2016) to the GAIL discriminator for improved regularization and has been successfully used for ILO with motion-capture data. To limit the effect of confounding factors during comparison, we share modules across AILO and the two baselines to the best of our ability. Concretely, all discriminator/reward networks use the same architecture and the gradient-penalty regularization (Mescheder et al., 2018) and thus exhibit the same reward biases. Furthermore, the MaxEnt-RL PPO module is the same for all algorithms.

**Performance.** Figures 2a–2c plot the learning curves for all the algorithms across the different tasks (heavy, drag, light). We show the average episodic returns achieved by the learner in the $l$-MDP, normalized to the returns achieved by the expert in $e$-MDP. The plots include the mean and the standard deviation of returns over 6 independent runs with random seeds.

We observe that AILO provides a noticeable improvement in learning efficiency in several situations, such as Half-Cheetah (heavy, drag, light), Walker (heavy), Hopper (heavy), Ant (drag), Humanoid (drag, light); while being comparable to the *best* baseline in other cases. For GAIfO, we find that it does not learn any useful skill for Walker (heavy) and exhibits training instability for Half-Cheetah (heavy, drag). We attribute this to the difficulty of matching the state-transition distributions across different dynamics. VAIL, which matches state distributions, instead of state-transition distributions, is a stronger baseline and works well in several cases. Lastly, we highlight a few failure modes of AILO – in Ant (light) and Humanoid (heavy), we note that AILO (and baselines) do not make much progress towards imitating the demonstrated behavior within our time budget, motivating the need for future enhancements to enable efficient skill transfer in these challenging setups.

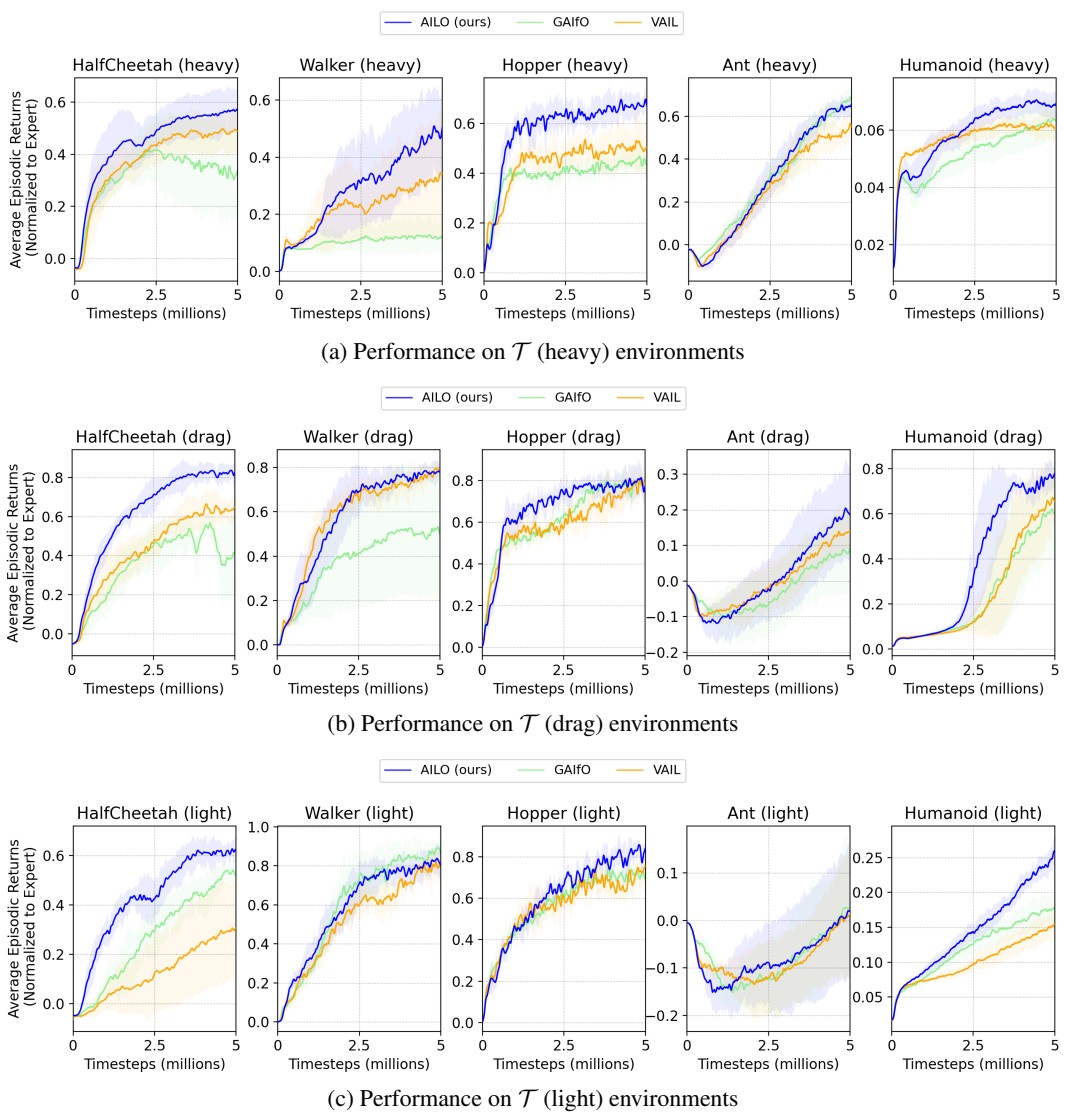

(a) Performance on $\mathcal{T}$ (heavy) environments

(b) Performance on $\mathcal{T}$ (drag) environments

(c) Performance on $\mathcal{T}$ (light) environments

Figure 2: Learning curves for AILO and the baselines for different environments with discrepancy in dynamics

**Ablation on the degree of dynamics mismatch.** For the empirical results in Figure 2, the variation in mass, gravity, or friction, between the $e$-MDP and the $l$-MDP was kept at a constant factor. In Figure 3, we consider the Walker task and plot the systematic degradation in the performance of AILO and the baseline GAIfO, as the $l$-MDP parameters drift further from the $e$-MDP parameters. We show the final average episodic returns achieved by the learner in the $l$-MDP, normalized to the returns achieved by the expert in $e$-MDP, as a function of the degree of dynamics mismatch. In the plots, the expert parameters (mass, friction) are kept constant and the learner parameters are varied such that the ratio increases from a starting value of 1 (no mismatch). We observe that although imitation naturally becomes more challenging as the dynamics become more different, the degradation with AILO is more graceful compared to GAIfO.

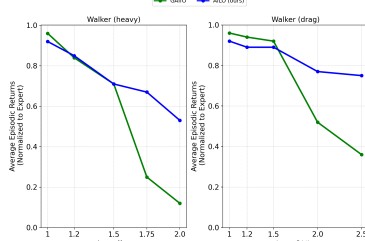

Figure 3: Results with different amount of dynamics mismatch

## 6 CONCLUSION

In this paper, we present AILO, our algorithm for imitation learning from observations under transition model disparity between the expert and the learner environments. Rather than directly matching the state-transition distributions across environments, we train an intermediary policy (advisor) in the learner environment and use it as a surrogate expert for the learner. Towards learning an advisor that acts as an effective surrogate, we propose to minimize the cross-entropy distance between the state-conditional next-state distributions of the advisor and the expert. To realize this idea into a scalable ILO algorithm, we leverage prior work on support estimation (RED). Our experiments on five MuJoCo locomotion tasks with different types of dynamics discrepancies show that AILO compares favorably to the baseline ILO methods in many cases.

## 7 REPRODUCIBILITY STATEMENT

The authors pledge to make the source code for reproducing all the experiments of this paper public upon the de-anonymization of the paper. The hyperparameters used for the results and some implementation details are included in Appendix A.2.

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

## A  APPENDIX

### A.1  IMPORTANCE SAMPLING FOR ADVISOR TRAINING

Following Eq. 5, the importance sampling factor to be estimated is $\frac{\rho_e(s)}{\rho_\pi(s)} \frac{\pi_a(a|s)}{\pi(a|s)}$, which is a product of two ratios. The second ratio $\frac{\pi_a(a|s)}{\pi(a|s)}$ is easily computable since we have both the advisor policy $(\pi_a)$ and the learner policy $(\pi)$ as parameterized Gaussian distributions. The first ratio $\frac{\rho_e(s)}{\rho_\pi(s)}$ can be approximated by training a binary classifier to distinguish the states sampled from the distributions $\rho_e(s)$ and $\rho_\pi(s)$. Concretely, consider the objective:

$$D' = \underset{D:\mathcal{S}\to(0,1)}{\arg\max} \; \mathbb{E}_{\rho_e(s)}\big[\log D(s)\big] + \mathbb{E}_{\rho_\pi(s)}\big[\log(1 - D(s))\big]$$

Then, we can estimate the ratio of the state distributions as $\frac{\rho_e(s)}{\rho_\pi(s)} \approx \frac{D'(s)}{1-D'(s)}$.

### A.2  HYPERPARAMETERS AND IMPLEMENTATION DETAILS

| Hyperparameter | Value |
|---|---|
| ***Discriminator / Reward network** ($f_\omega$)* | |
| architecture | 3 layers, 128 hidden-dim, tanh |
| # updates per iteration | 5 |
| optimizer | Adam (lr = $1 \times 10^{-4}$) |
| | |
| ***RL agent (shared)*** | |
| policy arch. | 3 layers, 64 hidden-dim, tanh |
| critic arch. | 3 layers, 64 hidden-dim, tanh |
| PPO clipping | 0.2 |
| PPO epochs per iteration | 5 |
| optimizer | Adam (lr = $1 \times 10^{-4}$) |
| discount factor ($\gamma$) | 0.99 |
| GAE factor (Schulman et al., 2015) | 0.95 |
| entropy target | $-|\mathcal{A}|$ |

Table 2: Hyperparameters for AILO and the baselines

The hyperparameters used for our experiments are mentioned in Table 2. The discriminator network used by the baselines and the reward network employed in AILO share a common structure. The same RL agent is used across AILO and all the baselines. Additionally, for the baselines, we tested the hyperparameters mentioned in the respective papers and tuned them with grid-search.

**Expert data collection.** We train the expert policy in $e$-MDP using environmental rewards with the PPO algorithm. We then generate a few rollouts with the trained policy and sub-sample state transitions from these rollouts. We create the expert dataset with 50 such transitions for {Half-Cheetah, Walker, Hopper, Ant}, and 1000 transitions for Humanoid.

**Absolute value for the expert scores.** In Figure 2, we plot the average episodic returns achieved by the learner in the $l$-MDP, normalized to the returns achieved by the expert in the $e$-MDP. For completeness, Table 3 reports the absolute value of the expected returns obtained by the expert per episode (max-episode-length = 1000 timesteps) in the $e$-MDP.

| Task | Returns |
|---|---|
| Half-Cheetah | 6441 |
| Walker | 4829 |
| Hopper | 3675 |
| Ant | 5765 |
| Humanoid | 9678 |

Table 3: Expert's performance in $e$-MDP

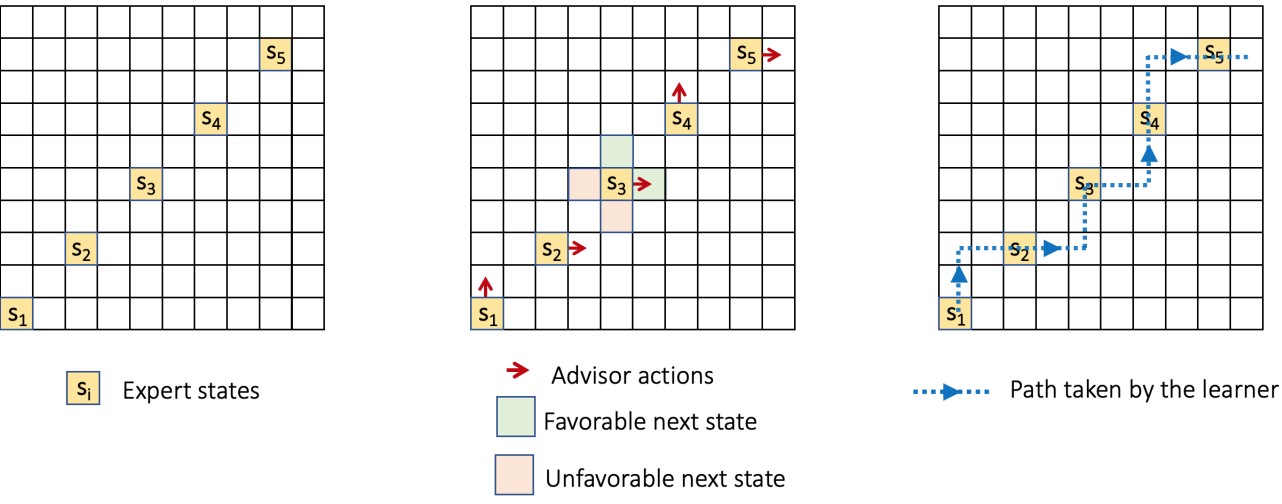

Figure 4: Illustration of the expert states, the actions by the advisor policy, and the path taken by the learner policy in a grid-world environment.

### A.3   ILLUSTRATION OF THE ADVISOR AND THE LEARNER IN GRID-WORLD

We discuss the differences between the advisor policy $\pi_a$ and the learner policy with the help of an illustration in the deterministic grid-world environment (Figure 4). The leftmost plot shows the demonstrated expert states $\{s_1, s_2, s_3, s_4, s_5\}$. The $e$-MDP and the $l$-MDP are the same grid-world but with the following transition dynamics mismatch – while the $e$-MDP allows diagonal hops such that the shown expert transitions are consecutive, in the $l$-MDP, the agent is restricted to only horizontal and vertical movements to the nearby cells.

As discussed in Section 3.2, we seek to train an advisor such that the dissimilarity between the destination state achieved with the expert policy in $e$-MDP and that achieved with the advisor in $l$-MDP is minimized. The middle plot in Figure 4 shows the actions (marked with red arrows) that an optimal deterministic advisor would take. Note that the advisor is trained *only* on the expert states to optimize for the next state; it is not trained for long-term behavior and thus struggles with decision-making on the non-expert states. At the state $s_3$, the green states are the favorable states (minimum distance to $s_4$), while the red states are the unfavorable states (maximum distance to $s_4$). In the $l$-MDP, the dataset $\{s_i, \pi_a(s_i)\}$ is then used to train a learner that runs in a closed feedback loop with the environment.

The learner is trained with the standard adversarial imitation learning objective to match its stationary discounted state-action distribution with that of the generated dataset $\{s_i, \pi_a(s_i)\}$ in the $l$-MDP. With adversarial imitation learning, the learner reproduces the demonstrated action at the demonstrated state, while also being trained to navigate back to the manifold of the demonstrated states when it encounters non-demonstrated states, thus enabling long-horizon imitation. The advisor, that is trained to optimize just the immediate action at each expert state (similar in principle to behavior cloning), is incapable of long-horizon imitation. The rightmost plot in Figure 4 shows the trajectory that an optimal learner may produce.

### A.4   COMPARING THE ADVISOR AND THE LEARNER PERFORMANCE

In Table 4, we report the average episodic returns achieved by the advisor and the learner in the $l$-MDP at the end of the training, normalized to the returns achieved by the expert in $e$-MDP. We show data for three environments and all the types of transition dynamics mismatch considered in the paper – *Heavy* (H), *Drag* (D), *Light* (L). We note that the learner outperforms the advisor by a substantial margin for all the scenarios. This is because, unlike the learner, the advisor is not optimized for long-horizon performance (*cf.* Appendix A.3).

|  |  | AILO-advisor | AILO-learner |
|---|---|---|---|
| HalfCheetah | (H) | 0.19 | 0.57 |
|  | (D) | 0.27 | 0.82 |
|  | (L) | 0.18 | 0.63 |
| Walker | (H) | 0.07 | 0.50 |
|  | (D) | 0.14 | 0.78 |
|  | (L) | 0.17 | 0.81 |
| Hopper | (H) | 0.53 | 0.67 |
|  | (D) | 0.48 | 0.77 |
|  | (L) | 0.66 | 0.84 |

Table 4: Normalized average episodic returns achieved by the advisor and the learner in the $l$-MDP.

## A.5 COMPARISON TO BEHAVIORAL CLONING AND RL BASELINES

In this section, we compare AILO with BCO (Torabi et al., 2018a). BCO is an ILO algorithm that first learns an inverse dynamics model $p(a|s, s')$ in the $l$-MDP, and then uses this model to infer the expert actions from the given expert state-transition dataset. These inferred actions are used to train the learner agent using the standard behavioral cloning strategy.

Table 5 reports the average episodic returns achieved by BCO and AILO, normalized to the returns achieved by the expert in $e$-MDP. We show data for all the environments and all the types of transition dynamics mismatch considered in the paper – *Heavy* (H), *Drag* (D), *Light* (L). We note that BCO performs much worse than AILO in all scenarios, with the exception of Ant (H). This can be attributed to two factors: 1.) the compounding errors problem in the behavioral cloning approach, and 2.) BCO does not explicitly account for the transition dynamics mismatch while inferring actions from the learned inverse dynamics model.

Although the agent does not receive rewards in the $l$-MDP in the imitation learning setup, we consider an oracle RL baseline that trains the agent to maximize the rewards in the $l$-MDP using the clipped-ratio PPO algorithm (Schulman et al., 2017). This baseline provides us with an estimate on the upper bound of the performance of AILO. Similar to the AILO numbers, the RL baseline scores are normalized to the returns achieved by the expert in $e$-MDP. The data in Table 5 shows that there is a lot of scope to optimize AILO further, especially for environments such as *Walker* and *Ant*.

|  |  | BCO | AILO | RL in $l$-MDP |
|---|---|---|---|---|
| HalfCheetah | (H) | $0.28 \pm 0.02$ | $0.57 \pm 0.08$ | 0.67 |
|  | (D) | $0.47 \pm 0.03$ | $0.82 \pm 0.04$ | 1.06 |
|  | (L) | $0.09 \pm 0.01$ | $0.63 \pm 0.05$ | 0.78 |
| Walker | (H) | $0.03 \pm 0.02$ | $0.50 \pm 0.14$ | 0.72 |
|  | (D) | $0.04 \pm 0.01$ | $0.78 \pm 0.03$ | 1.13 |
|  | (L) | $0.04 \pm 0.03$ | $0.81 \pm 0.08$ | 0.93 |
| Hopper | (H) | $0.10 \pm 0.01$ | $0.67 \pm 0.05$ | 0.65 |
|  | (D) | $0.09 \pm 0.01$ | $0.77 \pm 0.04$ | 0.76 |
|  | (L) | $0.11 \pm 0.05$ | $0.84 \pm 0.03$ | 0.86 |
| Ant | (H) | $0.73 \pm 0.01$ | $0.65 \pm 0.04$ | 1.04 |
|  | (D) | $0.02 \pm 0.00$ | $0.19 \pm 0.13$ | 0.83 |
|  | (L) | $0.06 \pm 0.02$ | $0.02 \pm 0.14$ | 0.75 |
| Humanoid | (H) | $0.02 \pm 0.00$ | $0.07 \pm 0.00$ | 0.48 |
|  | (D) | $0.01 \pm 0.00$ | $0.78 \pm 0.05$ | 0.71 |
|  | (L) | $0.01 \pm 0.00$ | $0.26 \pm 0.02$ | 0.32 |

Table 5: Normalized average episodic returns for BCO and RL with rewards in $l$-MDP.

## A.6 COMPARISON WITH I2L

In this section, we compare AILO with I2L  (Gangwani & Peng, 2020). I2L is an algorithm that builds on top of adversarial imitation learning and introduces specific components to handle the dy-

namics mismatch between the $e$-MDP and the $l$-MDP. AILO and I2L share the principle of curating an intermediate dataset in the $l$-MDP, which is then used to train the learner with adversarial imitation learning. In the case of I2L, the intermediate dataset is created by applying a filtration step on the trajectory generated in the $l$-MDP, by using a learned Wasserstein critic to rank the trajectories. Differently, in AILO, the intermediate dataset is obtained by the application of the advisor policy on the demonstrated expert states. These are two quite distinct and orthogonal approaches that could potentially be blended to create an effective intermediate dataset for the learner agent.

Table 6 reports the average episodic returns achieved by I2L, along with the performance of the other algorithms considered in this paper. We normalize all the scores to the returns achieved by the expert in $e$-MDP and highlight the best algorithm in each row with **bold**. The table verifies that I2L works very well for ILO under dynamics model disparity. Among the 15 environment-mismatch scenarios, GAIfO is the best method for 3, VAIL for 1, I2L for 5, and AILO for 7. This clearly indicates that a single approach may be insufficient to solve a large variety of ILO problems and opens up an interesting future direction of understanding the unique characteristics of these algorithms (*e.g.,* the contrasting approaches to create the intermediate dataset in I2L and AILO) towards designing efficient ILO algorithms.

| | | GAIfO | VAIL | I2L | AILO |
|---|---|---|---|---|---|
| HalfCheetah | (H) | $0.34 \pm 0.17$ | $0.49 \pm 0.09$ | $\mathbf{0.59 \pm 0.04}$ | $0.57 \pm 0.08$ |
| | (D) | $0.41 \pm 0.19$ | $0.65 \pm 0.07$ | $0.80 \pm 0.22$ | $\mathbf{0.82 \pm 0.04}$ |
| | (L) | $0.52 \pm 0.03$ | $0.31 \pm 0.20$ | $0.48 \pm 0.19$ | $\mathbf{0.63 \pm 0.05}$ |
| Walker | (H) | $0.13 \pm 0.07$ | $0.34 \pm 0.23$ | $\mathbf{0.56 \pm 0.09}$ | $0.50 \pm 0.14$ |
| | (D) | $0.50 \pm 0.29$ | $\mathbf{0.78 \pm 0.03}$ | $0.64 \pm 0.24$ | $\mathbf{0.78 \pm 0.03}$ |
| | (L) | $\mathbf{0.86 \pm 0.07}$ | $0.81 \pm 0.05$ | $0.76 \pm 0.05$ | $0.81 \pm 0.08$ |
| Hopper | (H) | $0.44 \pm 0.04$ | $0.50 \pm 0.10$ | $\mathbf{0.67 \pm 0.07}$ | $\mathbf{0.67 \pm 0.05}$ |
| | (D) | $\mathbf{0.79 \pm 0.03}$ | $0.78 \pm 0.04$ | $0.78 \pm 0.05$ | $0.77 \pm 0.04$ |
| | (L) | $0.70 \pm 0.06$ | $0.75 \pm 0.07$ | $0.70 \pm 0.08$ | $\mathbf{0.84 \pm 0.03}$ |
| Ant | (H) | $\mathbf{0.69 \pm 0.05}$ | $0.55 \pm 0.06$ | $0.48 \pm 0.22$ | $0.65 \pm 0.04$ |
| | (D) | $0.09 \pm 0.12$ | $0.14 \pm 0.06$ | $0.17 \pm 0.10$ | $\mathbf{0.19 \pm 0.13}$ |
| | (L) | $0.03 \pm 0.14$ | $0.02 \pm 0.14$ | $\mathbf{0.13 \pm 0.11}$ | $0.02 \pm 0.14$ |
| Humanoid | (H) | $0.06 \pm 0.01$ | $0.06 \pm 0.00$ | $0.07 \pm 0.01$ | $0.07 \pm 0.00$ |
| | (D) | $0.57 \pm 0.08$ | $0.65 \pm 0.10$ | $0.33 \pm 0.22$ | $\mathbf{0.78 \pm 0.05}$ |
| | (L) | $0.18 \pm 0.02$ | $0.15 \pm 0.02$ | $\mathbf{0.45 \pm 0.21}$ | $0.26 \pm 0.02$ |

Table 6: Normalized average episodic returns for different ILO algorithms.

