# OpenReview forum: "Imitation Learning from Observations under Transition Model Disparity"
_ICLR.cc/2022/Conference — ICLR 2022 Poster_

### Official Review · Reviewer_E4NG · 2021-10-29

**Correctness:** 3
**Technical Novelty And Significance:** 2
**Empirical Novelty And Significance:** 2
**Recommendation:** 6
**Confidence:** 4

**Main Review:**

The idea of using RED for estimating the joint density for consecutive states for imitation learning from observation is interesting, and the overall structure of the paper is clear and easy to follow.

Eq. (3) seems confusing: $J(\pi_a)$ is a function of $\pi_a$, but on the RHS of the equation it is taking a maximum over $\pi_a$. I am not sure what are taking maximum over since $\pi_a$ should be fixed in this situation as a input of $J(\pi_a)$. Also, the last equation does not hold if it is taking $\max$, it only hold if it is taking $\arg\max$. I do understand what the authors are trying to show here but this may need some revision. Similarly to Eq. (4).

Also the usage of the intermediate policy still lacks some intuition: if the advisor policy is able to recover the expert state trajectory, then optimizing the expected return s.t. the state distributions are the same should solve the problem. If the advisor policy is not able to fully recover the expert state trajectory, how would we expect the learner policy to recover the state distribution of the expert policy since the learner policy is matching the state-action distribution of the advisor policy (Eq. above Eq. (2))? For example, how is the learning path (the squiggly line) in Fig. 1 achieved by the proposed objectives?

The setting itself is not new: learning from expert and environments with different dynamics has also been proposed in many papers with different names: policy transfer, policy adaptation, domain transfer, sim2real, etc.

The main major issue is the experiment: the current baselines are weak since they are just LfO methods but do not specifically deal with dynamics mismatch. Even in this situation the proposed method is outperformed or evenly matched by the baselines. This is not the main issue, the main issue is the lack of comparison with baselines under the same setting as the proposed method, such as [1]. (Possibly, plus works in the above papers (policy transfer, policy adaptation, domain transfer, sim2real, ...) that work in the same setting.) Only comparing with baselines that do not specifically deal with dynamics mismatch does not seem very fair, and thus the experiment is not thorough enough for such an empirical paper.

[1] Tanmay Gangwani and Jian Peng. State-only imitation with transition dynamics mismatch. arXiv
preprint arXiv:2002.11879, 2020.

**Summary Of The Paper:**

The paper proposes a new algorithm for solving LfO task where the dynamics are different in the expert environment and the learner environment. The algorithm first trains an intermediate policy in the learner environment that mimics the expert's state trajectory and then the learner can just mimic this intermediate policy in the learner environment. The authors further show in the experiment that the proposed method outperforms previous LfO algorithms in several locomotion tasks.

**Summary Of The Review:**

Some mathematical representations still need some improvement. The experiment is not convincing enough. I would recommend a weak reject for the paper.

---

> ### Author Response · Authors · 2021-11-23
> **Response to the Reviewer E4NG**
>
> Thank you for your thoughtful comments and suggestions! We address the following concerns raised by the reviewer:
>
>
> **Intuition on the use of the advisor; is advisor policy able to recover the expert state trajectory; how is the squiggly path achieved in Fig 1:**
>
> The advisor policy does not recover the state trajectory of the expert. This is because the advisor is trained **only** on the expert states to optimize for the next state; it is not trained for long-term behavior and thus struggles with decision-making on the non-expert states. As such, running the advisor in a closed feedback loop with the environment results in the compounding error problem (as faced by the standard behavior cloning approach).
>
> We use the advisor to obtain the dataset {$s_i,\pi_a(s_i)$} in the $l$-MDP. We then train a learner policy with the adversarial imitation learning objective to match its stationary discounted state-action distribution with that of the generated dataset {$s_i,\pi_a(s_i)$} in the $l$-MDP. With adversarial imitation learning, the learner reproduces the demonstrated action at the demonstrated state, while also being trained to navigate back to the manifold of the demonstrated states when it encounters non-demonstrated states, thus enabling long-horizon imitation. The squiggly line in Fig. 1 reflects this capability of the learner to stay close to the expert states. This characteristic of adversarial imitation learning, and in general inverse-RL, is evident in practice and analyzed theoretically in a recent work [1].
>
> We have added Appendix A.3 that includes a visual example showing the expert states, the advisor, and the learner policies. In Appendix A.4, we report the performance of the trained advisor and the learner in several environments. We note that the learner outperforms the advisor by a substantial margin for all the scenarios. This is because, unlike the learner, the advisor is trained to optimize just the immediate action at each expert state (similar in principle to behavior cloning), and is therefore incapable of long-horizon imitation.
>
> We hope this provides a better intuition on the role of the advisor.
>
>
>
> **Correction in the mathematical representations:** Thank you for catching these. We have corrected Eq. 3 and Eq. 4 in the paper.
>
>
>
> **Comparison with I2L:** We have added Appendix A.6 comparing AILO with I2L [2]. In that section, we outline the similarities between the two algorithms and also the distinguishing features that set them apart. We observe that I2L works very well for ILO under dynamics model disparity.
>
> &nbsp;
>
> [1] Error Bounds of Imitating Policies and Environments, Xu et al.
>
> [2] State-only imitation with transition dynamics mismatch, Gangwani & Peng

---

> > ### Comment · Reviewer_E4NG · 2021-11-30
> > **Response to the Authors**
> >
> > Thank you for your update. I think Sec. A.4 has some convincing results, and I think the author's explanation in the rebuttal makes it more clear on the intuition on the usage of the advisor policy. However, I still find figure 4 not helpful enough: it's indeed more concrete than Fig. 1 (a) but they are essentially the same. I believe having the rebuttal contents incorporated in the main text would be very helpful for the readers to understand the intuition of the method. I also appreciate the new experiment results in App. 6. Since with the new contents and potential further improvements on the paper, I think in terms of the method itself it is clear and interesting, although the presentation may still require improvement. Overall I would like to raise my score to weak accept.

---

> > > ### Author Response · Authors · 2021-11-30
> > > **Thank you for the feedback!**
> > >
> > > We really appreciate the reviewer for taking our rebuttal into consideration and raising their rating of the paper. As suggested by the reviewer, we would blend the contents from the rebuttal (Appendix A.3 through A.6) into the main text to develop better intuition on the role of the advisor and improve the general presentation of the algorithm.

---

### Official Review · Reviewer_6eKr · 2021-10-29

**Correctness:** 3
**Technical Novelty And Significance:** 3
**Empirical Novelty And Significance:** 2
**Recommendation:** 6
**Confidence:** 4

**Main Review:**

### Summary++:
The setup for this paper is super interesting.  The domain is a realistic one: there is some limited data collected by simply watching an expert interact with the environment, but, where the learner is different in some critical way.  The question is then how do we leverage the observations of the expert to expedite learning an agent.  If solved generally (see 5 + 6 below) this would be a brilliant addition to the literature.

Sections two and four are _very_ well written.  Section one motivates the problem well, but doesn’t provide much detail on what content will be provided in the paper.  Section three is not well written, does not clearly explain the method, and is instead a bit of a drawn-out narrative that is hard to follow.  The empirical validation presented is lacking.  Finally, there is only a very short supplement that only provides hyperparameter / architectural settings and no further insight.

The crux of my critique is that I believe the empirical validation, to justify a method with this many moving parts, is lacking; and that the clumsy exposition of the methodology obfuscates why each component is necessary, and how they work in conjunction to create a performant policy.  While I believe there is a good method and a good conference submission in this line of work, it is not currently in that state.

### Major comments:
1 - My main complaint with this work is the lack of clarity in how the advisor actually works and what it provides.  The advisor is this magic box that encapsulates the expert policy _and_ the constraints/limitations of the learner.  The advisor is defined in learner space (as that is the only space we tractably have access to) and so how is the policy learned by the learner different to that of the advisor?  Why can’t the learner just imitate the advisor exactly (using, for instance, DAgger)?  If the learner can imitate the advisor exactly, then what is the point of the advisor?  Why not just learn the learner by whatever method you use to learn the advisor?  It is then also poorly explained how the learner is subsequently recovered from the expert demonstrations and the advisor policy.  Eq (2) is pointed to, but without much more detail as to the intuition that underlies this step.

2 - The method itself is also _very_ complicated.  As far as I can tell, there are at least four trained elements: the RED network, a reward network, an advisor policy and a learner policy.  That is a lot of stuff to specify, train, tune etc.  There is also no indication of how robust to hyperparameter/architecture settings the performance is.  i.e. If I happen to use a RED network with too-lower capacity, how is performance affected and how easy is it to diagnose that that is where the problem is?  I understand that all things RL are inherently a bit magic-boxey and can be sensitive, but even still, this method seems very convoluted/involved and is not sufficiently empirically validated to convince me that this complexity is warranted.

3 - Following on from 1 & 2, is the advisor and/or RED even strictly required?  It seems, and I may have gotten it wrong here, that you are essentially pre-training some policies to match the expert as well as it can, and then refining the policies in places where the expert can’t be matched.  I can just about believe that RED provides a tractable method for learning a policy that keeps as close to the experts state evolution as possible.  The advisor policy then steers the learner towards those actions?  Why not just weight the ILO loss by the prediction from RED?  Or just initialize the learner to match the state distribution where it can, and then refine based on reward?  Or something more akin to THOR [Sun, Bagnel & Boots, 2018]?  If the authors can comment on and clarify this.

4 - I will say upfront that this may be a little unfair as a critique, so take this with a pinch of salt.  The absolute performance of the method (and even the baselines) is disappointing.  Although I don’t know this for sure, I am relatively confident that something like regular SAC applied directly to the environment would achieve better performance in fewer environment interactions.  Therefore, why are we even bothering to use the expert demonstrations if regular RL in the learner could achieve competitive performance?  I was desperately hoping that this method would leverage the “imperfect” demonstrations to obtain a learner policy with dramatically fewer environment interactions, even if the absolute/final performance was slightly lower than the best-possible performance.  As I mentioned above, there is a TON of internal mechanics in this method, and while I believe there is an algorithm that can do ILO in this mismatched domain, I am as of yet unconvinced that the AILO method presented is _the_ method for doing it -- both in terms of absolute performance and technical overhead.

4.1 - The results in this paper have not convinced me that this method successfully, expediently and robustly extracts information from demonstrations from a slightly different environment to expedite learning a policy.

4.2 - I think the following “baselines” are missing:
- SAC/RL on the learner (also to provide the true upper bound on performance of the learner).
- DAgger on the learner using the misspecified expert policy.
- Behavioural cloning (BC) on just state-actions from the expert.
- Initializing a learner to the BC learned-policy and then performing RL to refine that policy.

Some of these are not practically applicable methods in real-life given the domain, but are, in my opinion, required to elucidate the performance of the method (bearing in mind that baselines are also there to help the _reader understand_ the performance and limitations of the method, as opposed to simply whether or not it works).

5 - I would also have liked to have seen the method applied to a smaller-scale toy example.  You even motivate the problem with a simple gridworld example -- why wasn’t the method applied to such a gridworld?  A simple example would help you pull apart the method and verify that all the components are working as intended, and would also allow instructive and intuitive visuals, graphs, diagrams, heatmaps etc to be generated to further help the reader _understand_ the method and its components, as opposed to simply posting a grid if graphs showing that it achieves half-way reasonable performance.

6 - I think there may also be a missed opportunity exploring (at least in principle) connecting IL and sim-to-real domains.  Sim-to-real seems to me to be a domain where this method (with some tweaking) could have some utility, as that is entirely derived from mismatch between two domains.  The advisor seems in-part to be a more flexible representation of an expert policy (learned in sim) that accommodates for the imperfections of the learner (learned at test-time in real).  There may be nothing in this, and it is just a suggestion, but the setups are so similar that _any_ comparison is conspicuous by absence -- even just commenting and discounting.

### Minor comments:

A - I would like to see a more specific explanation of the method earlier in the paper.  The advisor is loosely mooted early on, but is only really introduced with any specificity at the bottom of page 3.

B - Only proper nouns should be capitalized (“Reinforcement Learning” -> “reinforcement learning”;  “infinite horizon Markov Decision Process” -> “infinite horizon Markov decision process”).

C - Figure 1 should be improved to improve its clarity.  I have no idea what the “squiggly” lines represent.  Are these lines something we will learn?  The second part of that figure also offers no helpful content as it is too high-level.  I would also move it up to be a banner figure at the top of page two.  This would help frame your approach more succinctly and earlier on.  I would also consider (even more simply) pseudocoding the algorithm and placing it alongside the diagram on page two to drive home the method early.

D - Figure 2 could easily be compressed to save space if required.

E - The algorithm block is totally unhelpful.  It should either be made more precise using full mathematical notation, or, made more intuitive using pure pseudocode and high-level objects.

F - Releasing code for this is ultra important because I do not believe I could reproduce the results in this work from the prose and limited supplement alone.


**Summary Of The Paper:**

This paper tackles imitation learning from observations under model mismatch.  This is a realistic consideration when tackling more complex or real environments where methods such as IL (and its variants) may see dramatic improvements over more conventional RL-based approaches (to the point of being the only feasible approach).  Model mismatch makes existing ILO methods unstable (in theory, at the very least).  To ameliorate this, the paper introduces an _advisor_ policy, which learns from the expert demonstrations but under the constraints implicitly imposed by the capacity of the learner or the learners dynamics.  The advisor is then used to train the learner.  The method is empirically validated on some standard benchmarks and appears to perform reasonably well.


**Summary Of The Review:**

This work is interesting and shows promise, and if it were to work, would be a useful addition to the literature.  However, I am unconvinced by the method itself.  The method is quite involved, with lots of moving parts.  Furthermore, the presentation of the core of the method (the advisor) is remarkably poor, considering that it is _the_ contribution and that the rest of the paper is so well written.  The empirical validation is also a little lacklustre, both in terms of range and actual performance.  I also have more subjective issues with aspects of the paper (see comments above) that, while it would be unfair to reject solely on those issues, dissuade me from giving the authors the benefit of the doubt here.  Therefore, I recommend that this paper is not accepted for inclusion in this review cycle;  but note that with some reworking this paper could make a good submission to a good conference in the near future.

Good luck.

---

> ### Author Response · Authors · 2021-11-23
> **Response to the Reviewer 6eKr**
>
> Thank you for your thoughtful comments and suggestions! We address the following concerns:
>
>
> **Point #1: Main complaint: lack of clarity on the role of the advisor; how are the advisor and learner policies different; can learner use DAgger to learn from advisor:**
>
>
> The advisor and the learner policies are different in an important way. The advisor is trained **only** on the expert states to optimize for the next state (Equation 3); it is not trained for long-term behavior and thus struggles with decision-making on the non-expert states. As such, running the advisor in a closed feedback loop with the environment results in the compounding error problem (as faced by the standard behavior cloning approach).
>
> We use the advisor to obtain the dataset {$s_i,\pi_a(s_i)$} in the $l$-MDP. We then train a learner policy with the adversarial imitation learning objective to match its stationary discounted state-action distribution with that of the generated dataset {$s_i,\pi_a(s_i)$}  in the $l$-MDP. With adversarial imitation learning, the learner reproduces the demonstrated action at the demonstrated state, while also being trained to navigate back to the manifold of the demonstrated states when it encounters non-demonstrated states, thus enabling long-horizon imitation.  The squiggly line in Fig. 1 reflects this capability of the learner to stay close to the expert states. This characteristic of adversarial imitation learning, and in general inverse-RL, is evident in practice and analyzed theoretically in a recent work [1].
>
> To quantitatively support the above claim, we report the performance of the trained advisor and the learner in several environments. Appendix A.4 includes the data. We note that the learner outperforms the advisor by a substantial margin for all the scenarios.  This is because, unlike the learner, the advisor is trained to optimize just the immediate action at each expert state (similar in principle to behavior cloning), and is therefore incapable of long-horizon imitation.
>
>
> Regarding the question _“Why can’t the learner just imitate the advisor exactly (using, for instance, DAgger)”_: this is infeasible since the advisor cannot provide supervision on the states that are not included in the expert demonstration dataset.
>
>
> Lastly, we have added Appendix A.3 that includes a visual example showing the expert states, the advisor, and the learner policies. We hope this provides some clarity on the differences between the advisor and the learner, and the overall mechanics of the AILO algorithm. We’ll be happy to provide further clarifications.
>
>
>
>
> **Point #2: Method complicated; four trained elements; hyperparameter sensitivity:** Yes, our method has 4 elements to it -- {RED, reward-function, advisor, learner}. The RED network is pre-trained and kept fixed. The advisor policy is trained to optimize the immediate action at each expert state, thus reducing the learning complexity (compared to learning a long-horizon policy). The reward function and the learner policy components appear in almost all algorithms that are derived from GAIL, and several papers have introduced ideas to stabilize the learning dynamics. We understand the reviewer’s concern about the robustness of hyperparameter/architecture settings. In our empirical analysis, we found AILO to be very stable with configurations in the vicinity of those reported in Appendix A.2.
>
>
> **Point #3: Is the advisor and/or RED even strictly required?:** The importance of the advisor is that it helps to generate an intermediate dataset in the $l$-MDP that is effective for training the learner with adversarial imitation learning (please see the response to the first point). As for RED, we introduce it as a technique to approximate the expert’s state-transition distribution $\rho_e(s,s’)$ (required in Equation 3). It should be possible to substitute RED with other estimators and this is an interesting direction for future work.
>
>
> **Point #4: Missing baselines:** Given the time constraints, we could implement 2/4 new baselines that the reviewer mentioned. Please see Appendix A.5 for details.
>
> **Point #5: Intuitive visuals; understanding different components:** We have added Appendix A.3 that includes a visual example showing the expert states, the advisor, and the learner policies. Also, Table 1 in the paper provides insights on the validity of using the RED network as a substitute for the expert’s state-transition distribution. For the data in Table 1, we test the RED network with synthetic noise.
>
>
> **Point #6: Connections to Sim-to-real:** Thank you for the suggestion. You are absolutely right that there is a huge overlap between our setup and the sim-to-real problem. We will do a thorough literature survey and beef up the related works section.
>
>
> **What do “squiggly” lines represent in Fig 1.** Please see the response to the first point. We discuss this in that response.
>
> &nbsp;
>
> [1] Error Bounds of Imitating Policies and Environments, Xu et al.

---

> > ### Comment · Reviewer_6eKr · 2021-11-29
> > **Response**
> >
> > Dear authors,
> >
> > Thank you very much for engaging with my initial comments.  They have certainly allayed some of my concerns.  I still have reservations about the presentation, complexity, and the absolute "utility" of the method.  While I do not see these as sufficient grounds for rejection, as the work is technically sound an appears to work, they do prevent me from fully endorsing the paper for publication.
> >
> > Therefore, I will upgrade my score to a weak accept, although I request (it would be a requirement, if I there were such an option at this stage) that the authors further revising the manuscript to further simplify the exposition of the method.  These include:
> > - Downsizing and bringing the illustrative diagram in Supplemental Figure 4 into the main text.
> > - Clarifying the explanation of Figure 1.
> > - Reducing the size of the algorithm block and moving the comments inline.
> > - Reducing the bulk of text in the methods section (particularly when defining RED).
> > - Using the space this frees up to add a "TL;DR Section" explaining the method and technical hurdles before Preliminaries.
> >
> > Good luck.

---

> > > ### Author Response · Authors · 2021-11-30
> > > **Thank you for the feedback!**
> > >
> > > We really appreciate the reviewer for taking our rebuttal into consideration and raising their rating of the paper. We are also grateful to them for succinctly capturing the suggestions that would improve the exposition of the paper. The authors commit to making these changes to the final revision.

---

### Official Review · Reviewer_hbsF · 2021-11-02

**Correctness:** 4
**Technical Novelty And Significance:** 3
**Empirical Novelty And Significance:** 2
**Recommendation:** 5
**Confidence:** 3

**Main Review:**

$\textbf{Strengths}$:
- Empirically, they demonstrate their algorithm is more robust than baselines in the mismatch setting.
- The approach is quite innovative
- The problem they tackle is very interesting, as in realistic scenarios, their will usually be such dynamics mismatch.

$\textbf{Weaknesses}$:
- I found the paper slightly hard to parse, i.e. I had a hard time understanding for instance why we could not directly leverage the advisor policy, and why the learner policy trained so its occupancy would match the advisor's would be better suited than the advisor's. It would help if this could be further clarified, maybe with an intuitive visual example showing the expert, the advisor and the learner policies?


**Summary Of The Paper:**

This paper tackles the problem of imitation learning from observations only (no access to expert actions), in the setting where expert and agent are in different MDPs. More precisely, transition dynamics are different. The proposed approach consists in instantiating two agents, an advisor and a learner, such that the advisor is trained for its next state distribution to be close to the one of the expert, while the learner is trained so its state-action occupancy matches the one of the advisor.

The reason they instantiate this framework is that otherwise, next states may be unreachable by the agents hence the trained discriminator (if directly optimizing a GAIL loss between agent and expert occupancies) may perfectly classify and learning may then be challenging. Instead, in their setup the agent learns through a proxy living in the same MDP - the advisor.

Contributions:
- They introduce an algorithm aiming to mitigate the issue of agent and expert having dynamics mismatch
- Their demonstrate improved robustness to such mismatch in continuous control environments


**Summary Of The Review:**

I find the paper in this current form to be borderline due to the fact that it is hard to parse and to understand why the framework is instantiated in the way it is. It is still an interesting contributions, and empirical results are promising. Upon such clarifications to be made (along with potentially visual examples providing intuitions), I would be considering raising my score to a weak accept.

---

> ### Author Response · Authors · 2021-11-23
> **Response to the Reviewer hbsF**
>
> Thank you for your thoughtful comments and suggestions! We address the following concern raised by the reviewer:
>
> **Why we could not directly leverage the advisor policy, and why the learner policy trained so its occupancy would match the advisor's would be better suited than the advisor's:**
>
> The main reason why we can’t leverage the advisor directly is that the advisor is trained **only** on the expert states to optimize for the next state; it is not trained for long-term behavior and thus struggles with decision-making on the non-expert states. As such, running the advisor in a closed feedback loop with the environment results in the compounding error problem (as faced by the standard behavior cloning approach).
>
> We use the advisor to obtain the dataset {$s_i,\pi_a(s_i)$} in the $l$-MDP. We then train a learner policy with the adversarial imitation learning objective to match its stationary discounted state-action distribution with that of the generated dataset {$s_i,\pi_a(s_i)$} in the $l$-MDP. With adversarial imitation learning, the learner reproduces the demonstrated action at the demonstrated state, while also being trained to navigate back to the manifold of the demonstrated states when it encounters non-demonstrated states, thus enabling long-horizon imitation. The advisor, that is trained to optimize just the immediate action at each expert state (similar in principle to behavior cloning), is incapable of long-horizon imitation.
>
> To quantitatively support the above claim, we report the performance of the trained advisor and the learner in several environments. Appendix A.4 includes the data. We note that the learner outperforms the advisor by a substantial margin for all the scenarios. This is because, unlike the learner, the advisor is not optimized for long-horizon performance.
>
> Thank you for the suggestion of adding an illustration to convey this better. We have added Appendix A.3 that includes a visual example showing the expert states, the advisor, and the learner policies.

---

### Author Response · Authors · 2021-11-23
**Major revisions to the paper**

We sincerely thank the reviewers for spending the time to review the paper and provide quality feedback with constructive suggestions for improvement. We address each reviewer's comments separately below, and summarize the major revisions to the paper here:
* Added Appendix A.3 to better motivate the role of the advisor and illustrate the differences between the advisor and the learner
* Added Appendix A.4 to quantitatively measure the performance gap between the advisor and the learner in the $l$-MDP
* Added Appendix A.5, comparing AILO with a behavior-cloning based baseline and also an oracle RL baseline with rewards in the $l$-MDP
* Added Appendix A.6, comparing AILO with I2L [1]

&nbsp;

[1] State-only imitation with transition dynamics mismatch, Gangwani & Peng

---

### Decision · Program_Chairs · 2022-01-20

**Decision:**

Accept (Poster)

**Comment:**

The submitted paper considers the very interesting problem of imitation learning from observations under transition model disparity. The reviewers recommended 2x weak accept and 1x weak reject for the paper. Main concerns about the paper regarded clarity of the presentation, complicatedness of the proposed method, and experimental validation. During the discussion phase, the authors addressed some of the comments and provided an update of the paper providing additional details. While some of the reviewers' concerns still stand, I think the addressed problem is very relevant and the proposed method can be (with clarifications and improvements of the presentation) be interesting to parts of the community. Hence I am recommending acceptance of the paper. Nevertheless, I strongly urge the authors to carefully revise their paper, and taking the reviewers' concerns carefully into account when preparing the camera ready version of the paper.